# Abundance and diversity of resistomes differ between healthy human oral cavities and gut

Victoria R. Carr[1]*, Elizabeth A. Witherden[1], Sunjae Lee[1], Saeed Shoaie [1,2], Peter Mullany[3], Gordon B. Proctor [1], David Gomez-Cabrero[1,4,5,6] & David L. Moyes[1,6]*

The global threat of antimicrobial resistance has driven the use of high-throughput sequencing techniques to monitor the profile of resistance genes, known as the resistome, in microbial populations. The human oral cavity contains a poorly explored reservoir of these genes. Here we analyse and compare the resistome profiles of 788 oral cavities worldwide with paired stool metagenomes. We find country and body site-specific differences in the prevalence of antimicrobial resistance genes, classes and mechanisms in oral and stool samples. Within individuals, the highest abundances of antimicrobial resistance genes are found in the oral cavity, but the oral cavity contains a lower diversity of resistance genes compared to the gut. Additionally, co-occurrence analysis shows contrasting ARG-species associations between saliva and stool samples. Maintenance and persistence of antimicrobial resistance is likely to vary across different body sites. Thus, we highlight the importance of characterising the resistome across body sites to uncover the antimicrobial resistance potential in the human body.

---

[1] Centre for Host-Microbiome Interactions, Faculty of Dentistry, Oral and Craniofacial Sciences, King's College London, London, UK. [2] Science for Life Laboratory, KTH—Royal Institute of Technology, SE-171 21 Stockholm, Sweden. [3] Microbial Diseases, Eastman Dental Institute, University College London, London, UK. [4] Translational Bioinformatics Unit, NavarraBiomed, Departamento de Salud-Universidad Pública de Navarra, Pamplona, 31008 Navarra, Spain. [5] Unit for Computational Medicine, Karolinska Institutet (L8:05). Karolinska University Hospital, SE-171 76 Stockholm, Sweden. [6] These authors contributed equally: David Gomez-Cabrero, David L. Moyes. *email: victoria.carr@kcl.ac.uk; david.moyes@kcl.ac.uk

In recent years, antimicrobial resistance (AMR) has been highlighted as one of the biggest threats to global health, food production and economic development[1]. Given this rapidly developing global crisis, it is imperative that the current gaps in our understanding of the distribution, spread and associations of all AMR factors are filled. AMR is most often conferred through the expression of antimicrobial resistance genes (ARGs) that reduce a microbe's susceptibility to the effects of an antimicrobial compound. As such, monitoring the abundance and diversity of these ARG profiles, or the resistome, has huge potential to increase our understanding of the spread and persistence of AMR within a population. High-throughput next-generation sequencing technologies are beginning to be used as tools for screening ARGs for potential surveillance of antimicrobial resistance worldwide. Shotgun metagenomic data mapped against dedicated ARG reference databases are providing a wealth of insight into the resistomes of human[2–7] and animal guts[8,9], as well as the wider environment[10–13]. However, no large studies have, to date, attempted to characterise the resistome profiles of the human oral cavity. Commensal microbes from the oral cavity harbouring ARGs have potential to lead to antimicrobial resistant infections at other body sites. For example, β-lactam, clindamycin and erythromycin resistant strains of oral streptococci have caused infections at distal body sites such as infective endocarditis[14].

Metagenomic studies of the oral cavity indicate that this site potentially contains a diverse range of ARGs, including those encoding resistance to tetracycline, amoxycillin and gentamicin in saliva and plaque samples[15,16]. Thus, oral ARGs appear to be natural features of the human oral cavity. The presence of an oral resistome containing aminoglycoside, β-lactam, macrolide, phenicol and tetracycline ARGs in isolated Amerindian communities and ancient humans, indicates that the presence of these genes is not dependent on antibiotic exposure and is an inherent feature of the oral microbiome[17,18].

The oral microbial community faces unique ecological pressures, such as mechanical force, nutritional availability, pH levels, oxidative stress and redox potential. Despite these continually changing conditions, these communities have been shown to be relatively stable even after short-term antibiotic exposure. Horizontal gene transfer (HGT) has been documented as an important mechanism for the transfer and acquisition of ARGs within and between oral bacterial species[19,20]. The erythromycin resistance *mefA* and *mefE* genes have been found on the *MEGA* mobile genetic element associated with Tn*916*-like conjugative transposons (also called integrative conjugative elements ICE), and this has been implicated in conjugative transfer between viridans group streptococci (VGS) and other streptococci[21]. Thus, the oral microbiome contains a long-standing and mobile population of ARGs and is a significant reservoir for ARGs to be transferred to pathogenic microbes.

Here, we derive and compare the oral and the gut resistomes from 788 and 386 shotgun metagenomes, respectively, from healthy individuals from China[22], Fiji[23], the Philippines[24], Western Europe[25–27] and the USA[28]. We found country-specific differences in the proportion of saliva, dental plaque and stool samples containing ARGs, ARG classes and mechanisms. We made up to 415 comparisons of oral resistomes with paired gut resistomes derived from stool shotgun metagenomes from the same individuals, showing the oral resistome contains the highest and lowest abundances of ARGs, but a lower diversity of ARGs than the gut resistome. Overall, these results demonstrate the requirement for wider AMR surveillance studies at different body sites, including the oral cavity, to understand the composition of the resistome across different human microbial habitats.

## Results

**Country and body site-specific differences in resistomes.** To establish the incidence of ARGs in oral as well as stool metagenomes collected from various regions, metagenomes were mapped and quantified against the Comprehensive Antibiotic Resistance Database (CARD)[29]. Saliva samples were only available from China, Fiji, the Philippines and Western Europe. To account for the differences in read depths across different data sets, the samples were subsampled to the same number of reads across cohorts for absolute ARG incidence measures. The percentages of saliva samples that contain at least one ARG for each class and mechanism from these cohorts were evaluated. To account for varying read depth across samples, the samples were subsampled to the same number of sequences. Saliva samples from China, Fiji, the Philippines and Western Europe contain 20, 14, 23 and 17 ARG classes, respectively (Fig. 1a). Furthermore, ARG classes are found in Philippines saliva samples, but most of this variability originates from one individual: a farmer from Zambal who has carbapenem, fosfomycin, rifamycin and triclosan ARG classes[24]. All or almost all saliva samples from every cohort contain cephamycin, fluoroquinolone, lincosamide, macrolide, streptogramin and tetracycline ARGs, and a high percentage (above 50%) of saliva samples from all cohorts contain pleuromutilin ARGs. Unlike most cohorts, all saliva samples from China contain aminoglycoside ARGs represented by one ARG, *APH(3')-Ia*, and also a high proportion of these samples contain glycylcycline represented by one ARG, *tet(A)* (Supplementary Fig. 1a). The peptide ARG class is only found in saliva from Chinese and Philippines individuals. Mechanisms of antimicrobial resistance including antibiotic efflux, inactivation, target alteration and target protection are present in all saliva samples across all cohorts (Fig. 1b), whilst the antibiotic target replacement mechanism is found in China, Philippines and Western Europe, but not in Fiji. Reduced permeability to antibiotics is only found in saliva from the same farmer in Zambal.

Dental plaque metagenomic data were only available from China and the USA. The percentages of the China and USA plaque samples containing at least one ARG class and mechanism were compared, and found to consist of 16 and 18 ARG classes, respectively (Fig. 1c). A greater percentage of Chinese compared with USA plaque samples contain pleuromutilin and sulfonamide/sulfone ARGs. Similarly to saliva, all or almost all plaque samples from both cohorts contain lincosamide, macrolide, streptogramin and tetracycline ARGs, with a high percentage (above 50%) of these containing cephamycin, fluoroquinolone, glycylcycline and pleuromutilin ARGs. Notably, fluoroquinolone and tetracycline ARG classes in dental plaque are comprised fewer ARGs compared with saliva samples (Supplementary Fig. 1b). Antibiotic efflux, inactivation, target alteration, target protection and antibiotic target replacement mechanisms are present in all samples across both cohorts (Fig. 1d).

Stool samples were available from all locations, apart from the Philippines. Stool samples from China, Fiji, the USA and Western Europe were found to contain 31, 30, 17 and 30 ARG classes, respectively (Fig. 1e). All or almost all stool samples contain cephalosporin, cephamycin, diaminopyrimidine, lincosamide, macrolide, streptogramin and tetracycline ARGs, although most of these ARG classes are found in lower percentages of Fiji stool samples. Compared to oral samples, stool samples contain a lower proportion of the fluoroquinolone ARG class but higher proportions of cephalosporin and diaminopyrimidine ARG classes exclusively consisting of *CblA-1* and *dfrF* ARGs, respectively (Supplementary Fig. 1c). In addition to the USA stool samples containing the fewest number of ARG classes, they also contain a low proportion (less than 50%) of fluoroquinolones, penams (β-lactam with saturated five-membered ring, such

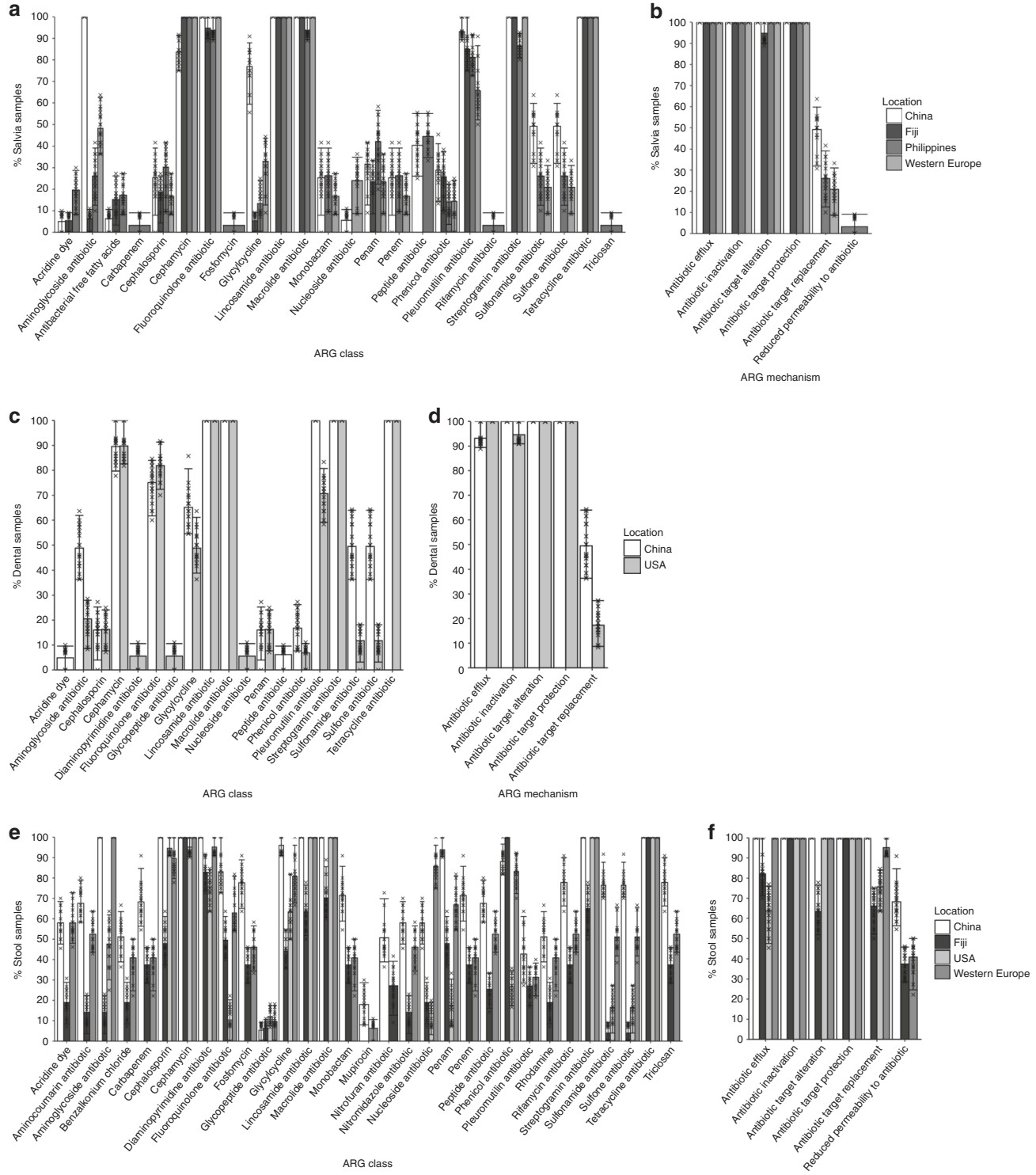

**Fig. 1 Percentage of individuals that contain ARG classes and ARG mechanisms.** Percentage of saliva samples that contain **a** an ARG class and **b** an ARG mechanism, of individuals from China ($n = 18$), Fiji ($n = 18$), the Philippines ($n = 18$) and Western Europe ($n = 18$). Percentage of dental plaque samples that contain **c** an ARG class and **d** an ARG mechanism, of individuals from China ($n = 18$) and the USA ($n = 18$). Percentage of stool samples that contain, **e** an ARG class and **f** an ARG mechanism, of individuals from China ($n = 18$), Fiji ($n = 18$), the USA ($n = 18$) and Western Europe ($n = 18$). The height of bars are the means and the error bars are 95% confidence intervals (CIs) of percentages extracted from bootstrapping samples 20 times shown by points. Source data are provided in the Source Data file.

as penicillin), penems (β-lactam with unsaturated five-membered ring), peptide and phenicols ARG classes compared to China, Fiji and Western Europe. Stool samples from China, Fiji and Western Europe contain resistance to triclosan, an antimicrobial that can

be found in many household cleaning products, with the highest proportion found in China. Antibiotic efflux, inactivation, target alteration, target protection and antibiotic target replacement mechanisms are present in all samples across all cohorts,

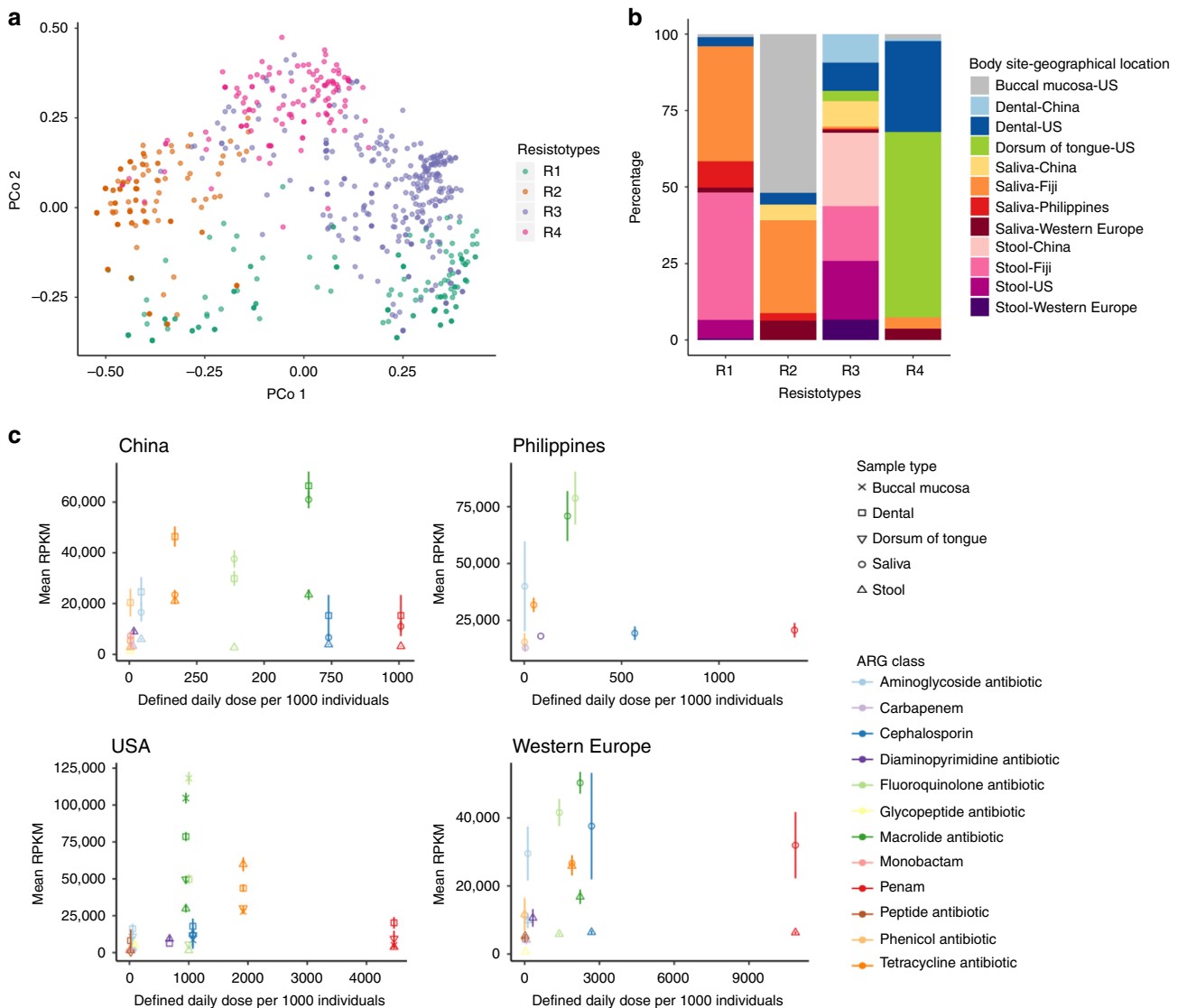

**Fig. 2 Clustering of ARG incidence profiles into distinct groups, and comparing ARG abundance to antibiotic use. a** Principal Coordinates Analysis of the incidence (presence/absence) of ARGs for all samples where each sample is represented by a point. Samples are labelled as Resistotype clusters, evaluated from hierarchical clustering of binary distance between ARG incidence profiles. The number of clusters was selected with the highest silhouette width using silhouette analysis. Samples from individuals from China (dental plaque: $n = 29$, saliva: $n = 33$, stool: $n = 72$), Fiji (saliva: $n = 129$, stool: $n = 136$), the Philippines (saliva: $n = 22$), the USA (buccal mucosa: $n = 86$, dental plaque: $n = 80$, dorsum of tongue: $n = 91$, stool: $n = 70$) and Western Europe (saliva: $n = 21$, stool: $n = 21$). **b** Percentage of resistotypes that contain samples from a body site and geographical location. **c** Mean and standard error (error bars) of reads per kilobase of read per million (RPKM) of ARGs for each ARG class against the defined daily doses per 1000 individuals in 2015 from China, the Philippines, Western Europe (France and Germany) and the USA. (Fiji antibiotic use data unavailable.) Mean RPKM calculated from individuals from China (dental plaque: $n = 32$, saliva: $n = 33$, stool: $n = 72$), Philippines (saliva: $n = 23$), USA (buccal mucosa: $n = 87$, dental plaque: $n = 90$, dorsum of tongue: $n = 91$, stool: $n = 70$) and Western Europe (saliva: $n = 21$, stool: $n = 21$). Source data are provided in the Source Data file.

but reduced permeability to antibiotics is not found in the USA (Fig. 1f).

To determine whether there are differences in ARG composition between oral and gut samples as well as between countries, the ARG incidence profiles for every sample were summarised using Principal Coordinates Analysis and clustered into distinct groups termed resistotypes. Resistotypes were identified using hierarchical clustering and silhouette analysis[30]. Four resistotypes in total were identified (Fig. 2a, b). Oral samples are mainly found in two major resistotypes, R2 and R4. R2 is comprised mainly buccal mucosa and saliva, and R4 contains mainly dental plaque and dorsum of tongue. All stool samples are found in only two major resistotypes, R1 and R3. R1 consists of mostly stool and

saliva from Fiji, whereas R3 contains mainly stool from China, Fiji, USA and Western Europe.

To evaluate whether the resistome is related to antibiotic prescription rates, the abundance of ARGs for every ARG class was compared with the defined daily doses per 1000 individuals of equivalent drug classes for each region. Prescription data were derived from ResistanceMap. This comparison indicates that overall ARG class abundance does not follow a significant linear relationship with antibiotic prescriptions for any country and body site (Fig. 2c; Supplementary Data 1).

The availability of longitudinal oral and stool samples from USA individuals who had not taken antimicrobial agents over two years afforded us the ability to investigate the stability of

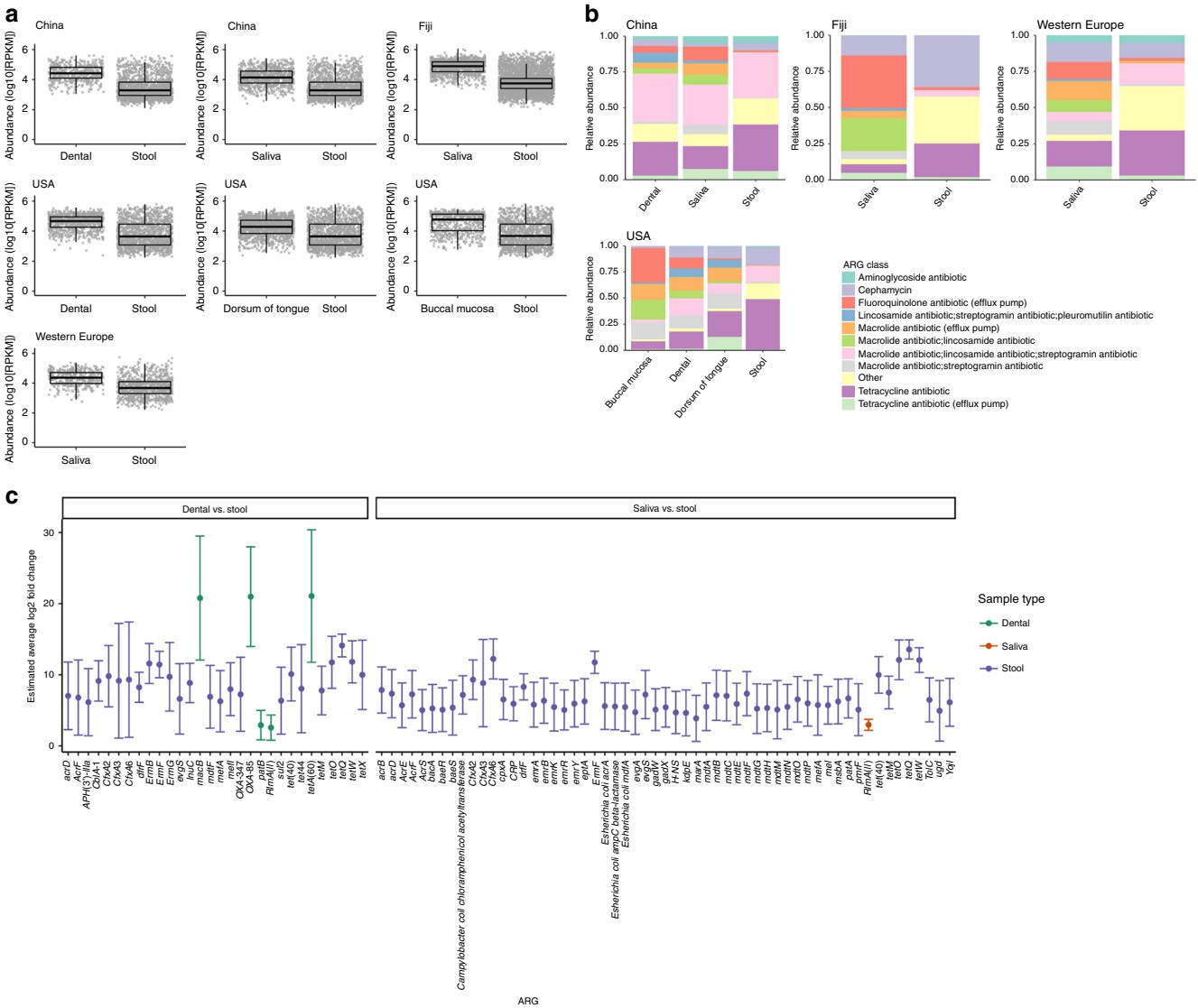

**Fig. 3 Comparing ARG abundance between the oral cavity and gut. a** Absolute abundance in log10 of reads per kilobase of read per million (RPKM) of ARGs for paired samples of individuals from China (stool and dental plaque: $n = 30$, stool and saliva: $n = 31$), Fiji (saliva and stool: $n = 132$), the USA (stool and dental plaque: $n = 68$, stool and dorsum of tongue: $n = 69$, stool and buccal mucosa: $n = 64$) and Western Europe (saliva and stool: $n = 21$). Centre line is median, box limits are upper and lower quartiles, whiskers are 1.5× interquartile ranges and points beyond whiskers are outliers. **b** Relative abundance of reads labelled by the top ten most abundant ARG classes across all geographical locations or other classes for each body site of individuals from China (dental plaque: $n = 32$, saliva: $n = 33$, stool: $n = 72$), Fiji (saliva: $n = 136$, stool: $n = 137$), the USA (buccal mucosa: $n = 87$, dental plaque: $n = 90$, dorsum of tongue: $n = 91$, stool: $n = 70$) and Western Europe (saliva: $n = 21$, stool: $n = 21$). **c** Estimated average log2 fold change of ARGs between paired dental plaque and stool, and saliva and stool samples using random effects meta-analysis across study cohorts ($p$-value < 0.05). Error bars are 95% confidence intervals from meta-analysis. ARGs selected for meta-analysis where adjusted $p$-value < 0.05 from differential abundance analysis between paired samples of individuals from China (stool and dental plaque: $n = 30$, stool and saliva: $n = 31$), Fiji (saliva and stool: $n = 132$), the USA (stool and dental plaque: $n = 68$) and Western Europe (saliva and stool: $n = 21$). Source data are provided in the Source Data file.

resistomes without antibiotics. Hierarchical clustering reveals that the same individuals and body sites cluster together, verifying that resistomes at all sites remain stable over a prolonged period with no antimicrobial selection pressure (Supplementary Fig. 2).

**ARG composition differs between the oral cavity and gut.** To further investigate the differences between oral and gut resistome profiles, the abundance and diversity between oral and gut samples from China, Fiji, USA and Western Europe were evaluated and compared. The total abundance, measured as the total reads per kilobase million (RPKM), of all ARGs in gut samples is lower than in oral (buccal mucosa, dental plaque, dorsum of tongue and saliva) samples across all pairwise comparisons in all

cohorts (Fig. 3a). The overall abundance is similar between paired US oral sites, with buccal mucosa and dental plaque having a slightly higher abundance than the tongue dorsum samples (Supplementary Fig. 3). Oral samples contain a higher relative abundance of ARGs coding for fluoroquinolone efflux pumps, lincosamide/streptogramin/pleuromutilin resistance, macrolide efflux pumps and macrolide/lincosamide resistance than stool samples (Fig. 3b; Supplementary Fig. 4). These classes are mostly dominated by one of two ARGs across all cohorts, such as *patB* (coding for part of the PatA-PatB efflux pump) in the fluoroquinolone efflux pump class (Supplementary Fig. 5a, b, c, d). Stool contains a higher proportion of tetracyclines and Other ARGs across all cohorts (Fig. 3b; Supplementary Fig. 4). These

Other ARGs are mostly found in aminocoumarins, aminoglycosides, cephalosporins, diaminopyrimidines, penams, penems and peptides across all cohorts (Supplementary Fig. 5a, b, c, d).

The abundance of individual ARGs were compared between sample types using differential analysis with DESeq2[31]. A meta-analysis strategy was implemented to combine results from all regions. Stool samples are enriched with more ARGs compared with oral samples, but oral samples have enriched ARGs of highest and lowest abundances compared with stool samples across all regions (Fig. 3c; Supplementary Fig. 6). *macB, OXA-85* and *tetA(60)* ARGs in dental plaque have the highest log fold changes (20.8, 21.0 and 21.1, respectively), whilst *patB* and *RlmA (II)* in plaque, and *RlmA(II)* in saliva, have the lowest log fold changes (2.9, 2.6 and 2.3, respectively) compared with stool samples (Fig. 3c). Highest log fold changes are seen in *cmlA6, Lactobacillus reuteri cat-TC, macB, TEM-1* and *tet32* from saliva (20.6, 20.5,19.8, 20.5 and 20.5, respectively), and *cmlA6, macB, OXA-85, pmrA, tetA(60)* and *tet(G)* from dental plaque (20.8, 20.6, 20.7, 20.7, 20.8 and 20.8, respectively) compared with stool samples in China that are not enriched across all cohorts (Supplementary Fig. 7). As well as differences between the oral and gut, differentially abundant ARGs were found between different sites in the oral cavity (Supplementary Fig. 6). For example, between the US dorsum of tongue and plaque samples, and between the US dorsum of tongue and buccal mucosa, all ARGs are enriched in the dorsum of the tongue. Similarly, between dental plaque and buccal mucosa, all ARGs are enriched in dental plaque. Most of these ARGs confer resistance to cephamycin, fluoroquinolone, MLS (macrolide/lincosamide/streptogramin) and tetracycline antibiotics. From China, there are more significantly abundant ARGs in saliva than plaque with resistance to aminoglycoside, cephalosporin, fluoroquinolone (*pmrA* and *patB*), lincosamides, macrolides (*macB* and *mefA* ARGs), MLS (in particular to the *Erm 235* ribosomal RNA methyltransferase family) and tetracycline antibiotics. Overall, stool samples are enriched with more alternative ARGs and ARG classes compared with oral samples, but with the highest and lowest enrichments of individual ARGs originating from oral samples.

To investigate ARG diversity further, the ARG richness was evaluated between pairwise comparisons of sample types for each cohort with ARG richness defined as the number of unique ARGs per sample. Although there are no significant difference between saliva and stool samples from Fiji and Western Europe, both Chinese and USA samples have significant differences in ARG richness. China and USA stool samples have a significantly higher ARG richness than paired China plaque and saliva, and paired USA plaque and buccal mucosa (Mann–Whitney, paired, two-sided *t* test, *p*-value < 0.05) (Fig. 4). In contrast, the USA dorsum of tongue contains a significantly higher ARG richness than USA stool. Between oral sites, Chinese saliva has a greater ARG richness than paired dental plaque (Mann–Whitney, paired, two-sided *t* test, *p*-value < 0.05). In addition, USA dorsum of tongue has a higher ARG richness than both plaque and buccal mucosa, whilst plaque has a greater ARG richness than buccal mucosa (Mann–Whitney, paired, two-sided *t* test, *p*-value < 0.05). It is important to note that while ARG richness only measures the gene incidence regardless of expression, multiple ARGs have the potential to be involved in the expression of a single efflux-pump complex, meaning ARG richness may overestimate this potential expression. Therefore, to determine the impact of this over-estimation, the analysis was repeated to exclude ARGs that regulate or are part of an efflux pump complex. The differences in ARG richness have the same significance across all paired samples and countries (Supplementary Fig. 8).

**Oral and gut ARG profiles associate with species.** Spearman's correlation analysis between ARG and species abundances were conducted to predict the origin of ARGs. Only significant correlations are found in saliva and stool from China, and saliva from the Philippines. The *CfxA* beta-lactamase family, *RlmA(II), tetQ, tetA(46), pgpB, patB* and *pmrA* ARGs are all strongly correlated with specific species found in both countries (Fig. 5a, b). The strongest co-occurrence can be found in saliva samples from China between *APH(3')-Ia* and a *Komagataella pastoris* strain, *Lactococcus lactis, Enterococcus faecalis* and *Leuconostoc mesenteroides*, whilst *pgpB* correlates with *Porphyromonas gingivalis*. The highly abundant ARG *RlmA(II)* in Chinese saliva is associated with *Gemella haemolysans, Veillonella parvula* and *Streptococcus mitis/oralis/pneumoniae*. In contrast to saliva, there are fewer species associated with a greater number of ARGs in stool samples from China. *E. coli* in Chinese stool samples is co-associated with many ARGs that encode multidrug efflux pumps and ARGs from *E. coli,* including *ampC beta-lactamase, acrA* and *mdfA* (Supplementary Fig. 9). Thus, this analysis has the potential to be a predictive tool of ARG origin in metagenomes. However, it can only be applied where an ARG or taxon is found in a high proportion (in this case, at least half) of the samples to ensure Spearman's correlation does not falsely rank many zero values.

**Discussion**

Antimicrobial resistance is one of the most serious health problems of recent times. The advent of high-throughput sequencing technologies has enabled us to analyse resistomes throughout a microbiome. In this study, we provide key insights into the resistomes of different intraoral sites from healthy individuals across diverse geographical locations and compare their composition to paired gut resistomes. At a population level, there are both country and body site-specific differences in the prevalence of ARGs, ARG classes and resistance mechanisms.. It is possible that differences in extraction protocols and batch effects may have a greater bias towards some ARGs over others. Therefore, we do not make direct statistical comparisons between cohorts. For China, the Philippines, USA and Western Europe, the abundance of ARG classes does not correlate with antibiotic prescription rates. A possible reason is for this is the prescription data does not include over-the-counter antimicrobial use, which is especially prevalent in China and the Philippines, and thus may underestimate antimicrobial use[32,33]. In addition, antibiotics are widely used in husbandry and the fishing industry with poorly understood impacts on AMR incidence and dynamics in humans[34–36]. Prescription levels for a particular antibiotic are unlikely to be of significant value in the surveillance of AMR in the regional community. Instead, determining the population resistome would be more informative[37].

The abundance and diversity of ARGs at different body sites is also of interest. Although, there are significantly more distinct ARGs in stool compared with oral samples, those ARGs present at the highest relative abundances exist in oral samples. There are several potential reasons why this may be. It is notable that many sites in the oral cavity (e.g., plaque and tongue dorsum) host highly complex and robust microbial biofilm structures. It has been posited that the compact structure of microbes within oral biofilms is a conducive environment for the aquistion of ARGs and their HGT within biofilms[38]. Likewise, the generally protective nature of biofilms against antimicrobial drugs may favour ARG acquisition. It is notable that the dorsum of tongue contains a higher diversity of these genes than other oral sites. This may be explained by the unique papillary structure on the dorsum of tongue which acts as complex microbiological niche favouring the deposition of oral debris and microbes[39], thus giving rise to a

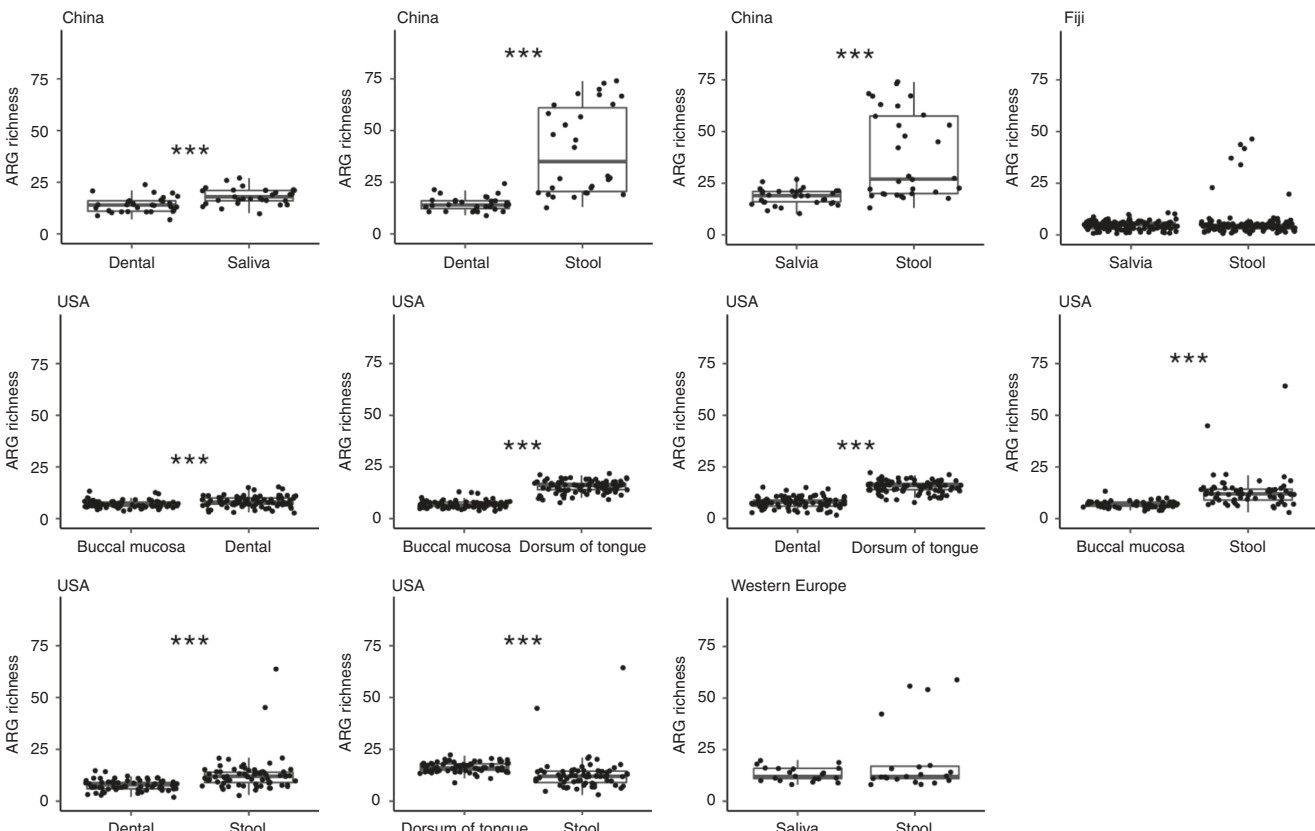

**Fig. 4 Comparing ARG richness between paired body sites.** ARG richness is defined as the number of unique ARGs for paired samples of individual from China (dental plaque and saliva: *n* = 31, stool and dental plaque: *n* = 30, stool and saliva: *n* = 31), Fiji (saliva and stool: *n* = 132), the USA (buccal mucosa and dental plaque: *n* = 78, buccal mucosa and dorsum of tongue: *n* = 86, dental plaque and dorsum of tongue: *n* = 89, stool and buccal mucosa: *n* = 64, stool and dental plaque: *n* = 68, stool and dorsum of tongue: *n* = 69) and Western Europe (saliva and stool: *n* = 21) with Mann–Whitney, paired, two-sided *t* test (*p*-value < 0.05 as *< 0.01 as **< 0.005 as ***). Centre line is median, box limits are upper and lower quartiles, whiskers are 1.5× interquartile ranges and points beyond whiskers are outliers. Source data are provided in the Source Data file.

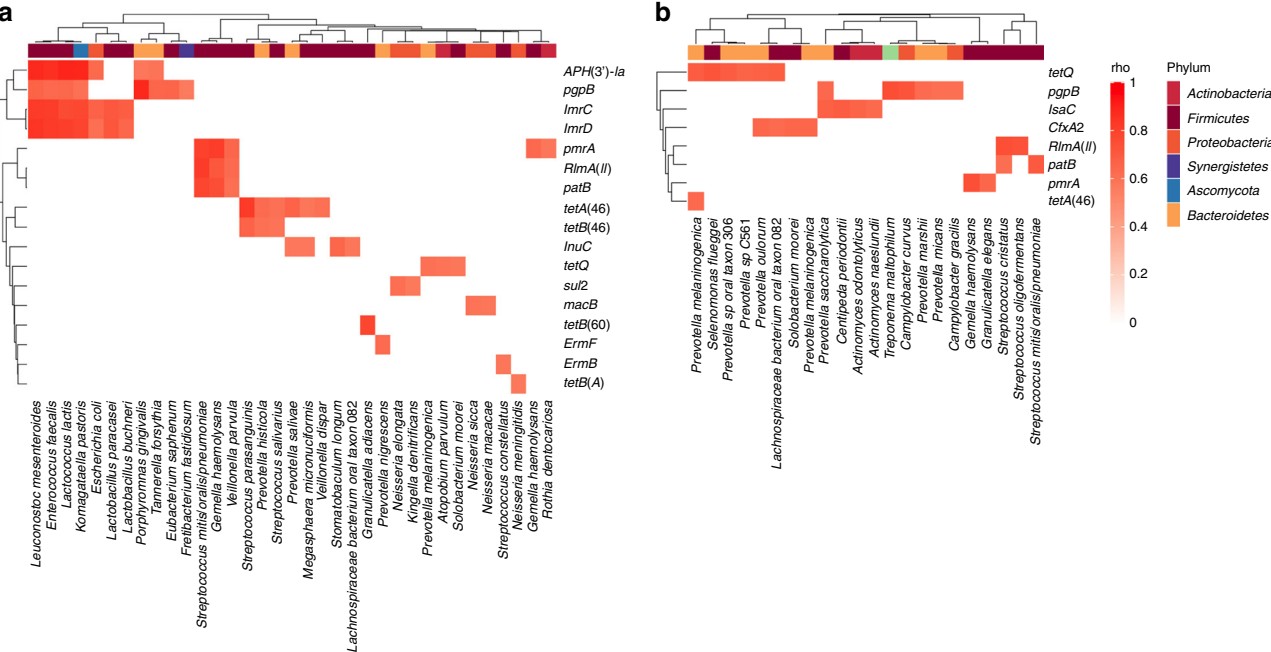

**Fig. 5 Spearman's correlation of ARG and species abundance from saliva samples.** Each heatmap represents correlations of individuals from **a** China (*n* = 31) and **b** Philippines (*n* = 23). Rows and columns are clustered by hierarchical clustering of Euclidean distance. Columns are coloured by phylum. *P*-values are adjusted by Benjamini–Hochberg multiple test correction. *Rho* shown only where adjusted *p*-value < 0.05. Source data are provided in the Source Data file.

richer microbial community with potentially greater numbers of transient microbes. Another reason could be the difference in species resident in the gut compared with the oral cavity. *E. coli* strains in Chinese stool samples are predicted to contain a variety of ARGs, especially of the multidrug class, whereas species found in Chinese saliva were estimated to contain fewer ARGs. Due to stringent constraints of the correlation analysis, however, it was not possible to predict the origins for ARGs for all cohorts.

Antibiotic use leading to acquisition of ARGs is another potential factor. Pharmacokinetics of orally administered antibiotics suggests that the oral cavity and oesophagus would be only briefly exposed to the antibiotic during swallowing, whilst the gut is exposed over a more prolonged period. As antibiotics transit through the intestinal tract, they are gradually absorbed via the intestinal epithelium into the bloodstream. Therefore, microbes in the gut exposed to antibiotics for a longer period of time due to their increased bioavailability will receive higher antibiotic selection pressures than those in the oral cavity[40]. The incidences where the oral cavity is likely to acquire ARGs from selective pressures of local antibiotics are from topical antibiotics for periodontal infections or orally administered antibiotics being absorbed into the bloodstream.

The differences in ARG profiles across body sites has significant implications for the characterisation and interpretation of resistome studies. Previous shotgun metagenomic studies have focused almost exclusively on the resistome from the human gut[2–4,6]. Whilst the gut may be a diverse reservoir of ARGs, whether these genes are particularly prevalent or have the potential for expression sufficient to drive resistant infections at other body sites is not clear[41]. It is therefore imperative that to test potential applications of non-culture-based metagenomic AMR surveillance, we need to characterise the resistome at different body sites with different pharmacokinetic exposures to antimicrobials. This information can then be integrated with culture-based susceptibility tests, culturomics[42] and functional metagenomic screens[43] to determine the expression potential of these ARGs. In doing so, we will obtain a more complete picture of the state of AMR within a population.

## Methods

**Metagenomic sequence data**. A total of 1174 publicly available metagenomic samples covering the USA, China, Fiji, the Philippines and Western Europe (France and Germany), all sequenced using Illumina HiSeq 2000, were analysed. Longitudinal USA samples were excluded from the majority of the study after the first time point to ensure each sample was independent, unless specified otherwise. All metagenomes passed over half the quality control metrics in FastQC 0.11.3 (https://www.bioinformatics.babraham.ac.uk/projects/fastqc/) with these pass rates calculated in MultiQC[44]. These samples include (1) longitudinal data across two years with various timepoints from the Human Microbiome Project 1 (referred to as USA)[28] containing buccal mucosa ($n = 87$: 32 with one, 36 with two, 18 with three and 1 with six timepoints); dorsum of tongue ($n = 91$: 22 with one, 43 with two, 24 with three and 2 with four timepoints); dental plaque ($n = 90$: 23 with one, 43 with two, 20 with three, 1 with four and 3 with six timepoints); stool ($n = 70$: 13 with one, 33 with two, 21 with three, 2 with four and 1 with six timepoints), (2) healthy control samples from a Chinese rheumatoid arthritis study[22] containing dental plaque ($n = 32$); saliva ($n = 33$); stool ($n = 72$), (3) saliva ($n = 136$) and stool ($n = 137$) samples from Fiji[23], (4) saliva samples ($n = 23$) from healthy hunter-gatherers and traditional farmers from the Philippines[24] and (5) saliva ($n = 21$) and stool ($n = 21$) samples from Western Europe (5 saliva and 5 stool samples from Germany[25,27], and 16 saliva and 16 stool samples from France[26,27]).

Raw paired-end metagenomic reads from Chinese and Philippines samples were downloaded from the EBI. Paired-end metagenomic samples from USA were downloaded from https://portal.hmpdacc.org/. Raw paired-end metagenomic reads from Fiji (project accession PRJNA217052), France and Germany (project accession PRJEB28422) were downloaded from the NCBI. All USA, China, Fiji and Philippines samples, and stool samples from France and Germany, were collected and sequenced as described in the following cited studies[22–26,28]. Saliva samples from France and Germany were collected and sequenced as described in the following cited study[27]. Metadata for the samples can be found in Supplementary Data 2.

**Processing metagenomic data**. The raw reads for all samples were trimmed using AlienTrimmer 0.4.0[45] with parameters *-k 10 -l 45 -m 5 -p 40 -q 20* and Illumina contaminant oligonucleotides (https://gitlab.pasteur.fr/aghozlan/shaman_bioblend/blob/18a17dbb44cece4a8320cce8184adb9966583aaa/alienTrimmerPF8contaminants.fasta). Human contaminant sequences were removed from all samples by discarding reads that mapped against a human reference genome (downloaded from Human Genome Resources at NCBI on 27th February 2017) using Bowtie2 2.2.3[46] with parameters *-q -N 1 -k 1 --fr --end-to-end --phred33 --very-sensitive --no-discordant*. The quality of the raw reads and the filtered reads of each sample was evaluated using the FastQC 0.11.3.

**Identifying ARGs**. All processed metagenomes were mapped against *nucleotide_fasta_protein_homolog_model* from the antimicrobial resistance database CARD 3.0.0[29] using KMA 1.2.6. Hits were identified where the template coverage was greater than 90%. The metagenomes were mapped against these hits using Bowtie2 2.2.5 with parameter *–very-sensitive-local*. Mapped reads were filtered from unmapped reads, sorted and indexed using Samtools 1.9[47]. Statistics for the number of reads mapped for each ARG were identified using Bedtools 2.28.0[48].

**Abundance of ARGs**. The reads per kilobase of read per million (RPKM) was calculated for every sample as the number of reads divided by the total number of library reads per million, then divided by the gene length in kilobases. The relative abundance of ARGs for each country and sample type was calculated by dividing the RPKM by the sum of RPKM for each country and sample type. The relative abundance of ARGs for each sample and sample type was calculated by dividing the RPKM by the sum of the RPKM for each sample. Differential abundance of ARGs between paired sample types from each country were calculated using the DESeq2 1.20.0 package[31], as recommended by Jonsson et al.[49]. ARGs that were significantly differentially abundant (adjusted *p*-value < 0.05) across study cohorts for paired sample comparisons were identified using a meta-analysis random effects model with the metafor 2.1-0 package[50].

**ARG class abundance and antibiotic prescription rate**. The mean RPKM and standard error for every ARG class was compared against the antibiotic prescription rate measured as the defined daily dose (DDD) per 1000 individuals in 2015 from China, the Philippines, Western Europe (France and Germany) and the USA. This data was derived from ResistanceMap, accessed on 19th February 2019. No antibiotic use data was available for Fiji. Linear regression was conducted on the log transformed mean RPKM versus the DDD Per 1000 for all ARG classes.

**Percentage of samples with ARGs, ARG classes and mechanisms**. To show whether the percentages of samples containing an ARG class were consistent across the same number of reads, metagenomes were first subsampled using seqtk 1.2 with parameter seed *-s100*. 6.9 million reads were subsampled from 18 saliva samples with the lowest number of reads >6.9 million reads, from each cohort: China, Fiji, the Philippines and Western Europe. 18 million reads were subsampled from 18 dental plaque with the lowest number of reads greater than 18 million reads from both China and USA cohorts. 16.9 million reads were subsampled from 18 stool samples with the lowest number of reads greater than 16.9 million reads from China, Fiji, the USA and Western Europe cohorts. These were mapped to CARD 3.0.0 as described in *Identifying ARGs*. R 3.5.1 was used for all downstream analysis. Each ARG was annotated with Drug Class and Resistance Mechanism using CARD 3.0.0 metadata. Percentages of samples containing an ARG, ARG class and mechanism were calculated from these samples. Ninety-five percent confidence intervals (CIs) were evaluated from percentages identified from bootstrapping samples 20 times for each cohort and sample type.

**Principal coordinates analysis**. In all, 790 metagenomes that contained at least 1 millions reads and were not longitudinal USA samples were first subsampled to 1 million reads using seqtk 1.2 with parameter seed *-s100*. These were mapped to CARD 3.0.0 as described in *Identifying ARGs*. Principal Coordinates Analysis was applied to the binary distance between ARG presence or absence profiles for each sample (excluding longitudinal US samples) using the vegan 2.5-2 package (https://cran.r-project.org/web/packages/vegan/index.html). Resistotypes were identified using hierarchical clustering of the Euclidean distance between principal coordinates with eigenvalues above zero. Silhouette analysis was used to determine the optimum number of resistotypes using the cluster 2.0.7.1 package (https://cran.r-project.org/web/packages/cluster/index.html). The number of resistotypes is defined by the number of clusters with the largest silhouette width.

**ARG diversity**. To ensure the ARG richness could be compared statistically across different sample types from the same individuals[51], the metagenomes (excluding longitudinal USA samples) were subsampled using seqtk 1.2 with seed *-s100*. Paired samples from the same individuals in each of the following groups containing two sample types were subsampled to a number rounded down by two significant figures from the lowest number of reads in the group.

*China dental plaque vs. saliva:* 3.5 million reads were sampled from China dental plaque ($n = 31$) and paired saliva ($n = 31$) samples. *China stool vs. saliva:* 3.5 million reads were sampled from China stool ($n = 31$) and paired saliva ($n = 31$) samples. *China stool vs. dental plaque:* 14 million reads were sampled from China stool ($n = 30$) and paired dental plaque ($n = 30$) samples. *USA buccal mucosa vs. dental plaque:* 1 million reads were sampled from USA buccal mucosa ($n = 78$) and paired dental plaque ($n = 78$) samples. *USA buccal mucosa vs. dorsum of tongue:* 1 million reads were sampled from USA buccal mucosa ($n = 86$) and paired dorsum of tongue ($n = 86$) samples. *USA buccal mucosa vs. stool:* 1 million reads were sampled from and USA buccal mucosa ($n = 64$) and paired stool ($n = 64$) samples. *USA dental plaque vs. dorsum of tongue:* 4.2 million reads were sampled from USA dental plaque ($n = 89$) and paired dorsum of tongue ($n = 89$) samples. *USA dental plaque vs. stool:* 4.2 million reads were sampled from USA dental plaque ($n = 68$) and paired stool ($n = 68$) samples. *USA dorsum of tongue vs. stool:* 14 million reads were sampled from USA dorsum of tongue ($n = 67$) and paired stool ($n = 67$) samples. *Fiji saliva vs. stool:* 1.2 million reads were sampled from Fiji saliva ($n = 132$) and paired stool ($n = 132$) samples. Fiji samples containing fewer than 1.2 million reads were excluded from the analysis. *Western Europe saliva vs. stool:* 3.1 million reads were sampled from Western Europe saliva ($n = 21$: 5 from Germany and 16 from France) and paired stool ($n = 20$: 5 from Germany and 16 from France) samples. These were mapped to the CARD database as described in Methods Identifying *ARGs*.

Once the metagenomes were subsampled, ARGs identified and filtering by coverage, the ARG diversity per sample was measured as the ARG richness, recommended previously by Bengtsson-Palme et al.[52]. For every sample, the ARG richness was calculated as the number of unique ARGs. To account for multiple ARGs coding for an efflux pump complex, the ARG richness was calculated excluding ARGs that regulate or are part of an efflux pump complex. The ARG richness between samples in each group was tested for statistical significance with a Mann–Whitney, paired, two-sided $t$ test.

**Correlation analysis**. MetaPhlAn2 2.6.0[53] was used to identify taxonomic composition from all samples. Spearman's correlation was applied to relative abundances of reads mapped to ARG and MetaPhlAn2 species profiles for paired samples. ARGs and species that were not found in more than half of samples for each country were removed, to alleviate the bias from potential joint ranking of zero values by Spearman's rank. The rho and *p*-values were calculated using the *stats* package in R, and the *p*-values were adjusted with Benjamini–Hochberg where FDR < 5%. Correlations were found from China saliva and paired stool samples, and Philippines saliva samples. No significant correlations could be found from Fiji, Western Europe or US samples.

**Reporting summary**. Further information on research design is available in the Nature Research Reporting Summary linked to this article.

## Data availability
ARG data, figures and tables are available at https://github.com/blue-moon22/resistomeData. Data underlying Figs. 1–5 and Supplementary Fig. 1–9 are provided in the Source Data file. All other data are available from the corresponding author upon reasonable requests.

## Code availability
R package for resistome analysis is available at https://github.com/blue-moon22/resistomeAnalysis. The script to run the analysis is available at https://github.com/blue-moon22/resistomeData.

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

## Acknowledgements

The research was supported by the Centre for Host-Microbiome Interactions, King's College London, the Biotechnology and Biological Sciences Research Council (BBSRC) (Grant No. BB/M009513/1) awarded to D.L.M., and Engineering and Physical Sciences Research Council (EPSRC), BBSRC and UK Research and Innovation (UKRI) (Project No. EP/S001301/1) awarded to S.S. The authors would like to thank Professor William Wade for constructive criticism of the paper. S.S. acknowledges Uppmax for providing assistance for computational infrastructure.

## Author contributions

V.C. and D.L.M. conceived the presented idea. V.C. and D.G-C. conducted the bioinformatics and data analysis. V.C. and D.L.M. wrote the paper with support from D.G-C., E.W., G.B.P., P.M., S.L. and S.S. D.L.M. and D.G-C. oversaw the project.

## Competing interests

The authors declare no competing interests.
