## [Peer Review File · Nature Communications]

Reviewers' comments:

Reviewer #1 (Remarks to the Author):

Peer Review of "Abundance and diversity of resistomes differ between healthy human oral cavities and gut"

This study by Carr et al is a survey of putative antibiotic resistance gene (ARG) abundance in human oral cavities and guts using previously published metagenomic datasets from the US, China, Fiji, the Philippines, and Western Europe. The authors used BLAST to compare their reads to the Comprehensive Antibiotic Resistance Database (CARD) to identify ARGs. The authors state that oral resistomes have lower ARG richness but higher ARG abundance than gut resistomes. They additionally performed in silico phage gene detection, principal coordinate analysis, and correlation analysis to determine which genes and species co-occurred with resistance genes.

Characterization of the relatively understudied oral resistome and its comparison to the gut resistome is of scientific and clinical interest. The authors' use of several geographically distinct human cohorts has the potential to provide a global perspective on this topic. Their large study size can enable a robust and statistically powered analysis of the differences in ARGs between the oral and gut resistomes. Regrettably, the methods described to address these questions appear sufficiently flawed and/or poorly described to make it impossible to evaluate the validity of most of the claims herein. The method described for identifying ARGs in sequencing reads is flawed in that it does not appear to control for read length, normalize for the length of ARG sequences, and most importantly has no consideration of the false positive rates that are commonplace in numerous other published tools for similar analyses. Indeed, there are several software packages that are designed for this type of analysis (e.g. KmerResistance, SRST2, ARIBA, AMRFinder, and ShortBRED) that would provide better normalization and more accurate ARG identification, and the authors provide no justification for why they use a new methodology for this purpose with virtually no benchmarking or comparisons to existing tools. Furthermore, while CARD is an excellent, well-curated database, using it in the way described can result in erroneously high antibiotic resistance abundance and diversity, since by default it includes regulatory elements and multi-component efflux pumps that require several proteins to confer resistance. Compounding the methodological problems, the manuscript writing is unclear at times, with numerous grammatical errors and disorganized arguments. Even if the results were taken at face value, the discussion requires more fleshing out to biologically contextualize the various trends reported, with clear consideration of previously published work. The discussion currently contains unfounded speculation, for instance regarding ARG 'transcription' (this is a DNA-only based study) and horizontal gene transfer. While no new data or validated methods are offered in this manuscript, an in-depth survey of ARGs between body sites (esp. understudied ones like the oral microbiome) in publicly available datasets would undoubtedly add scientific value. As is, I cannot recommend consideration of this manuscript in its

current form, given that a considerable portion of the analysis is inherently flawed. I offer the following major and minor critiques to guide the authors in reconsidering their rather interesting questions with substantially reworked analyses:

Major critiques:

1) With metagenomic sequencing, more reads will map to longer genes. This is accounted for in a publicly available ARG quantification software, such as those listed above. The authors do not report doing this at any step, which confounds any further analysis.

2) The alignment methods used in this paper to identify ARGs allows for a high false discovery rate that can be remedied by using software packages designed for this type of analysis. Further, their description of the cut-offs used (lines 356-7) doesn't adequately describe any percent identity thresholds, which could result in more false-positives.

3) The different methods of gene identification used are also worrisome. Phage genes are identified by annotating contigs using Prodigal and using BLAST to identify hits from identified genes. ARGs, on the other hand, are identified by mapping reads to the ARG using Bowtie2. No justification is provided for why these different methods are used.

4) Different versions of CARD are used for the analyses in this paper. Reads are initially mapped to version 2.0.0 (line 353) but then annotated using version 2.0.3 (line 358) and finally in the phage portion of their paper they used version 3.0.0 (line 369). No justifications are provided for use of these different versions.

5) The authors have performed no statistical tests on their pairwise ARG abundance comparisons (Fig. 3a), or else all comparisons are insignificant and major findings need to be revised.

6) One explanation for increased diversity of ARGs in stool was the enrichment of *E. coli* in gut microbiomes compared to oral cavities. This finding was not validated in other cohorts, and *E. coli* is only one of many species that are known for carrying diverse ARGs. This conclusion needs to be walked back or supported in other cohorts.

7) Another explanation that needs revision is that the gut microbiome has low ARG abundance but high ARG diversity which "may be the result of some of the highly diverse ARG population only being present in species that are unable to transcribe them and are therefore irrelevant." Since their analysis does not include any RNA data they can draw no conclusions about ARG transcription, and it appears that the authors may have confused ARG abundance in the DNA metagenomes with RNA expression level.

8) Making comparisons between datasets that were previously sequenced with different methods has many risks, not limited to batch effects and extraction bias. A discussion of the limitations of interpretability of between-country and between-datasets comparison is needed.

a) Further, datasets often have different read lengths as sequencing technology has changed. To compare across studies read length needs to be normalized. There is no discussion of read length and the parameters used in their trimming would not appear to correct for this.

b) The same issues very much apply to detecting phage content in metagenomes- it is highly extraction method dependent, and can accordingly lead to dramatically different estimates of phages in the extracted and sequenced DNA.

c) To offset the above, the authors need to be very careful with claims about the differences between countries

9) The identification of ARGs on contigs with phage genes is not confirmation of horizontal gene transfer. Be very careful with language regarding this, especially in the discussion (line 270).

Minor critiques:

1) Line 72 sentence “relative to their use” please edit to clarify that use is antibiotic prescription rates in the country

2) General writing tip: please, check for, unnecessary commas, between sentence, clauses. They make, things, quite difficult to, read. (sorry for being facetious here).

3) Introduction, last paragraph: transfer some of the results summary to abstract rather than the introduction

4) Euclidean should be capitalized throughout.

5) Figures from page 31 on are low-res and unreadable. Figures need to be made much larger.

6) Line 305 spelling: “focusses”

7) The classes of antibiotics chosen to be displayed in Fig 3 are odd and not comprehensive or reflective of antibiotic use patterns or previously reported abundant ARG classes. Beta-lactamases, one of the most common and widely disseminated classes of ARGs, is not mentioned in this paper.

8) Line 159: it appears “containments” should be “contaminants”.

9) The language needs to be cleaned up in the paragraph on differential analysis (lines 205 – 227). The purpose of this analysis is not to determine which ARGs each sample “contains” but rather the ARGs that are differentially “enriched” in each sample type. Sample composition was already covered in the first paragraph of the results section.

10) Line 407: it appears “where” should be “were”.

11) Fig. 5; Supplementary Fig. 9 captions: “Bonferroni-Hochberg” should either be Bonferroni or Benjamini-Hochberg.

12) Supplementary figure numbering in the text appears to be wrong in some instances.

a) Lines 189 and 192: “Supplementary Fig. 5” appears to refer to Supplementary Fig. 4.

b) Line 207 “Supplementary Fig. 7” appears to refer to Supplementary Fig. 6.

c) Supplementary Fig. 7 is not referenced in the text.

13) When multiple panels showing the same measurement are included in the same panel the y-axis should be the same to avoid misrepresentation of the data (Figs. 3a, 4; Supplementary Figs. 3, 6, 8.)

Reviewer #2 (Remarks to the Author):

Increasing antibiotic resistances are a global burden. The authors analysed the oral and the gut resistomes from 790 + 386 public available metagenomes. They found significantly different resistomes in both organs; the oral resistome contains a higher abundance but a lower diversity of ARGs than the gut resistome. Moreover, they describe country-specific resistomes, which correlate with the antibiotic spectrum therapeutically available.

Novelty.

The authors pronounce that this is the first study that characterises the oral cavity resistome worldwide. In contrast, country-specific antibiotic use practices impact the human gut resistome (Genome Res 2013) were already described.

Methods.

Public available shotgun metagenomes were analysed; however, the sampling design of the studies analysed was different and the samples are very heterogeneous. In some cases, low number of individuals representing one country, e.g. five from Germany.

It is briefly mentioned that the quality of all datasets was checked following the standardized filtering. However, there is no information available on the quality requirements that were used for this study. Were datasets with lower quality removed from the analysis or manually filtered in order to meet the requirements?

Only one method was applied: in silico analyses of metagenomes. For risk assessments of antibiotic resistances it is much better to combine various methods and set everything in a specific context.

The authors mention (when referring to previous studies) that individuals that were not exposed to anthropogenic sources of therapeutic antibiotics also harbor diverse ARGs in their microbiome. This is later on mostly ignored when the authors draw conclusions from their own results.

In order to better understand the relevance of the findings it would be necessary to correlate the data with more precise metadata, which indicates which individuals were exposed to certain therapeutic antibiotics.

The metagenomic data should be ideally linked to other -omics data (metagenomics or – transcriptomics) to assess the expression of ARGs, otherwise it can't be excluded that inactive artifacts are assessed.

Reviewer #3 (Remarks to the Author):

This is a well-conducted well-written study and I basically have not comments regarding the methodologies, the presentation or the technical details.

My only concern is whether the data from the different studies are comparable. Thus, we do know that different DNA-purification methods can drastically change the composition of the microbiome, including the resistome. Thus, some evidence that the differences observed is not only due to the data generation would be nice.

I am a bit surprised with the correlation between antimicrobial use and resistance. This is in some contrast to recent studies looking at both metagenomics and single isolates (see Collignon. *lancet Planetary health* and hendriksen *Nature Commun.*).

I only have some problems identifying the real novelty of the study and the purpose of doing it. The main findings seem to be that the oral cavity contains more resistance and less diversity compared to gut microbiome and then that some abundances of bacteriophages are associated to some resistance genes. Considering the very large number of resistance genes they looked at the latter is not surprising and it would have been nice with some verification using long-read sequencing (I know that would require new samples) or at least binning to show co-localisation.

Reviewer #1:

This study by Carr et al is a survey of putative antibiotic resistance gene (ARG) abundance in human oral cavities and guts using previously published metagenomic datasets from the US, China, Fiji, the Philippines, and Western Europe. The authors used BLAST to compare their reads to the Comprehensive Antibiotic Resistance Database (CARD) to identify ARGs.

For most of our diversity and abundance analysis, we used Bowtie2. We only used BLASTp to identify where phages and ARGs co-localise.

The authors state that oral resistomes have lower ARG richness but higher ARG abundance than gut resistomes. They additionally performed in silico phage gene detection, principal coordinate analysis, and correlation analysis to determine which genes and species co-occurred with resistance genes.

Characterization of the relatively understudied oral resistome and its comparison to the gut resistome is of scientific and clinical interest. The authors' use of several geographically distinct human cohorts has the potential to provide a global perspective on this topic. Their large study size can enable a robust and statistically powered analysis of the differences in ARGs between the oral and gut resistomes.

Regrettably, the methods described to address these questions appear sufficiently flawed and/or poorly described to make it impossible to evaluate the validity of most of the claims herein. The method described for identifying ARGs in sequencing reads is flawed in that it does not appear to control for read length, normalize for the length of ARG sequences, and most importantly has no consideration of the false positive rates that are commonplace in numerous other published tools for similar analyses. Indeed, there are several software packages that are designed for this type of analysis (e.g. KmerResistance, SRST2, ARIBA, AMRFinder, and ShortBRED) that would provide better normalization and more accurate ARG identification, and the authors provide no justification for why they use a new methodology for this purpose with virtually no benchmarking or comparisons to existing tools. Furthermore, while CARD is an excellent, well-curated database, using it in the way described can result in erroneously high antibiotic resistance abundance and diversity, since by default it includes regulatory elements and multi-component efflux pumps that require several proteins to confer resistance. Compounding the methodological problems, the manuscript writing is unclear at times, with numerous grammatical errors and disorganized arguments. Even if the results were taken at face value, the discussion requires more fleshing out to biologically contextualize the various trends reported, with clear consideration of previously published work. The discussion currently contains unfounded speculation, for instance regarding ARG 'transcription' (this is a DNA-only based study) and horizontal gene transfer. While no new data or validated methods are offered in this manuscript, an in-depth survey of ARGs between body sites (esp. understudied ones like the oral microbiome) in publicly available datasets would undoubtedly add scientific value. As is, I cannot recommend consideration of this manuscript in its current form, given that a considerable portion of the analysis is inherently flawed. I offer the following major and minor critiques to guide the authors in reconsidering their rather interesting questions with substantially reworked analyses:

1. We would like to thank the reviewer for suggesting alternative tools to consider as part of our analysis. We had previously assessed the variability of some of these tools, whilst we have subsequently assessed the others as listed below:
 - AMRFinder can only be applied assemblies rather than reads as it uses BLAST and Hidden Markov Model searches only suitable for longer sequences
 - ShortBRED is a tool for extracting marker amino acid sequences from a protein reference database and proteins of interest for alignment with sequencing data.

For ShortBRED, we would not have been able to generate a comprehensive collection of proteins of interest. To attempt this, we would have assembled all our metagenomes and identify proteins (using Prodigal or Plass). These assemblies will likely lose information between highly homologous ARGs leading to the generation of incomplete and erroneous marker sequences.

- SRST2 shares many similarities with our pipeline. It is a Bowtie2-based mapping tool with allele typing. The authors of this tool do not recommend using allele typing for metagenomes as SRST2 is built to detect “one specific allele” in a Whole Genome Sequence (WGS) of a single isolate, rather than many alleles in a whole metagenome. However, the author suggests detecting the depth of the genes using gene typing only (<https://github.com/katholt/srst2/issues/90>). Their gene typing approach uses Bowtie2 and Samtools with a 90% coverage threshold, the same as we use in our paper, although SRST2 uses older versions than those we have used. We would also like to highlight that other recent resistome metagenomics papers have also used similar mapping and coverage threshold approaches using Bowtie2, indicating that this a tried, tested and widely accepted method (Hendriksen et al., 2019; Pärnänen et al., 2018; Zaheer et al., 2018).
- We have also considered the k-mer-based tools ARIBA and KmerResistance. The authors of ARIBA advise against using their tool for metagenomic datasets (Hunt et al., 2017) but authors of KmerResistance suggest this tool could be used for metagenomic datasets (Clausen et al., 2016). KmerResistance performs both mapping to an antimicrobial resistance database and species prediction. The mapping part of KmerResistance is dependent on a k-mer based scoring algorithm called KMA which aligns reads more accurately than Bowtie2 against ARG databases of high redundancy in sequence composition (Clausen et al., 2018) and a recent preprint has used KMA to identify ARGs in in metagenomic data (Sturød et al., 2019). We found that the CARD database contains some redundancy which we had not considered in our original methods, and we therefore conducted clustering of CARD sequences with greater than 90% identity using CD-HIT finding 2238 sequences could be binned into 938 clusters. As a result, we have now updated our pipeline to use KMA to account for redundancy in the CARD database with a 90% coverage threshold. We then use Bowtie2 to map the metagenomic reads against the ARGs identified in KMA to extract the read counts for RPKM normalisation. Hereafter we refer to this process as the KMA-Bowtie2 pipeline in our responses.
- We conducted a comparison of our previous method using Bowtie2-only, and alternative methods SRST2, KMA-only and our new approach of using KMA-Bowtie2, all with a 90% coverage threshold. All tools show a very similar distribution of relative abundances of ARG classes (Figure 1 below) and a similar distribution of ARGs (figures available on request). The main differences are in the relative abundances of the lincosamide ARG class (higher in KMA-only), whereas lincosamide/streptogramin/pleuromutilin ARG class is higher in other tools. This figure has not been included in our current revised manuscript, but can be added if the reviewer feels this is appropriate.

2. The reviewer raises a concern that regulatory elements and multi-complex efflux pumps could lead to erroneously high antibiotic resistance abundance and diversity measures. For abundance measures, we have updated the legends in Figure 3b and Supplementary Figure 4 to include efflux pumps labels for clarity. In addition, we label ARGs as being “part of” or “regulate” named efflux pumps in the volcano plots of shown in Supplementary Figure 6 of the manuscript. For diversity measures, we provide a Supplementary Figure 8 of the ARG richness excluding ARGs regulating or part of an efflux pump. The statistical significance was found to be the same as in Figure 4 of ARG richness with all ARGs.
3. Other concerns are addressed under each point made by the reviewer below.

Major critiques:

1) With metagenomic sequencing, more reads will map to longer genes. This is accounted for in a publicly available ARG quantification software, such as those listed above. The authors do not report doing this at any step, which confounds any further analysis.

We apologise for not explaining our methods more clearly in controlling for read length and normalisation of ARG length. To quantify ARGs, we used Bowtie2 in our original manuscript to extract the read counts. The reads counts are normalised by length using RPKM. This has originated from RNA-Seq studies and is commonly applied to metagenomic studies. This has been defined more clearly in the methods under a separate section titled “Abundance of ARGs” (Line 498 of the revised manuscript).

2) The alignment methods used in this paper to identify ARGs allows for a high false discovery rate that can be remedied by using software packages designed for this type of analysis. Further, their description of the cut-offs used (lines 356-7) doesn't adequately describe any percent identity thresholds, which could result in more false-positives.

In our original manuscript we use a coverage threshold of 90% from our Bowtie2-only method. In our current revised manuscript, we continue to use a coverage threshold of 90% for our new KMA-Bowtie2 pipeline. The 90% coverage threshold was selected by benchmarking read coverage of ARGs from our new pipeline against ARGs identified from metagenomic assemblies using the method depicted below in Figure 2. Generally, the coverage threshold also excludes ARGs with a percentage identity under ~92 % so the benchmarked 90 % coverage threshold is adequate.

3) The different methods of gene identification used are also worrisome. Phage genes are identified by annotating contigs using Prodigal and using BLAST to identify hits from identified genes. ARGs, on the other hand, are identified by mapping reads to the ARG using Bowtie2. No justification is provided for why these different methods are used.

In the original manuscript, both phages and ARGs were identified with BLASTn for the comparative study. This was stated in the Methods (Lines 379 – 385 previous manuscript). However, we have now removed this analysis in our revised manuscript as we didn't feel the existence of phage proteins was sufficient evidence to suggest ARGs putatively associated with mobile phages (i.e. ancestral bacterial DNA could have acquired phage proteins through integration of prophages that became immobile)

4) Different versions of CARD are used for the analyses in this paper. Reads are initially mapped to version 2.0.0 (line 353) but then annotated using version 2.0.3 (line 358) and finally in the phage

portion of their paper they used version 3.0.0 (line 369). No justifications are provided for use of these different versions.

We would like to thank the reviewer for pointing out this inconsistency and oversight. We realised we used different versions of CARD throughout the process of analysing data for the different tests while preparing the manuscript. Although we believe that there were few changes that could have affected our results between the different versions, we cannot rule out there may be inconsistencies. Therefore, we have re-analysed the data using CARD 3.0.0 in all cases to ensure consistency.

5) The authors have performed no statistical tests on their pairwise ARG abundance comparisons (Fig. 3a), or else all comparisons are insignificant and major findings need to be revised.

We understand the concern for not performing a statistical test. However, we did not feel it was appropriate conducting a statistical test between the ARG abundance as it represents the sum of the RPKM per sample and does not distinguish differences in individual ARG abundances. Therefore, we have added an additional Fig. 3c summarising the differential abundance analysis showing the meta-analysis of the log₂ fold change in ARG abundance between individual ARGs.

6) One explanation for increased diversity of ARGs in stool was the enrichment of *E. coli* in gut microbiomes compared to oral cavities. This finding was not validated in other cohorts, and *E. coli* is only one of many species that are known for carrying diverse ARGs. This conclusion needs to be walked back or supported in other cohorts.

Regarding our assertion that *E. coli* are predicted to contain a variety of ARGs, we could only find this significant correlation between bacteria and ARG abundances in Chinese stool samples (Lines 289-290 previous manuscript). We could not see any significant correlations in other cohorts due to our strict correlation criteria stated in our Methods (Lines 451 – 454 previous manuscript). We would like to apologise for our oversight in generalising this result and have reworded our conclusion in the revised manuscript to clarify this is only relevant to Chinese samples only (Lines 414 – 416 revised manuscript).

7) Another explanation that needs revision is that the gut microbiome has low ARG abundance but high ARG diversity which “may be the result of some of the highly diverse ARG population only being present in species that are unable to transcribe them and are therefore irrelevant.” Since their analysis does not include any RNA data they can draw no conclusions about ARG transcription, and it appears that the authors may have confused ARG abundance in the DNA metagenomes with RNA expression level.

We thank the review for pointing out this potential source of confusion and this statement has now been removed from our revised manuscript.

8) Making comparisons between datasets that were previously sequenced with different methods has many risks, not limited to batch effects and extraction bias. A discussion of the limitations of interpretability of between-country and between-datasets comparison is needed.

The review raises a good point here. The potential impact of batch effects and extraction bias have now been discussed in more detail in the Discussion. We are very aware of this issue, and therefore we only compared cohorts on high-level data abstraction, i.e. the proportion of samples containing ARG classes and mechanisms in Figure 1 rather than comparing individual measures on the abundance and diversity of ARGs (for example) across cohorts. We have found that saliva and buccal mucosa samples have a lower read depth than other oral sites. This could be a result of an underestimation of absolute values

(ARG incidence and ARG richness) in these samples. To mitigate this, the reads were subsampled so they had the same number of reads for direct comparisons of absolute values. Throughout our analysis we are careful to avoid statistical tests across cohorts and focus on statistical comparisons between paired body sites within the same cohorts. As such, differences across body-sites within the same individual, have been subjected to the same methodological/sequencing techniques, aiding in comparability.

a) Further, datasets often have different read lengths as sequencing technology has changed. To compare across studies read length needs to be normalized. There is no discussion of read length and the parameters used in their trimming would not appear to correct for this.

This is a valid point for these types of studies. However, all cohorts included here were sequenced using an Illumina HiSeq 2000 machine, meaning that differing read length or sequencing technology differences are not an issue in this instance. A statement indicating this has now been added to the Methods.

b) The same issues very much apply to detecting phage content in metagenomes- it is highly extraction method dependent, and can accordingly lead to dramatically different estimates of phages in the extracted and sequenced DNA.

This is an important point in studies including analyses of phages. As a result, in the original manuscript, we did not make direct inferences in the text on comparing phage profiles across cohorts. However, as stated above we have now removed the phage analysis in our revised manuscript as we did not feel this added sufficient narrative to our study.

c) To offset the above, the authors need to be very careful with claims about the differences between countries

We agree and as a result have removed claims we made in the first section of results in comparing countries related to antimicrobial use based on Figure 1.

9) The identification of ARGs on contigs with phage genes is not confirmation of horizontal gene transfer. Be very careful with language regarding this, especially in the discussion (line 270).

We agree absolutely with the reviewer, and have removed this claim from our revised manuscript.

Minor critiques:

1) Line 72 sentence “relative to their use” please edit to clarify that use is antibiotic prescription rates in the country

Changed to “prescription rates”

2) General writing tip: please, check for, unnecessary commas, between sentence, clauses. They make, things, quite difficult to, read. (sorry for being facetious here).

This has been rectified.

3) Introduction, last paragraph: transfer some of the results summary to abstract rather than the introduction

We have transferred some of the results summary from the last paragraph of the Introduction to the Abstract, specifically greater similarity in interpersonal resistomes between the same body sites (Lines 29 – 30 of the revised manuscript). We have also added

that abundance antimicrobial resistance classes does not correlate with antibiotic prescription rates (Lines 26 – 27 of the revised manuscript).

4) Euclidean should be capitalized throughout.

Changed

5) Figures from page 31 on are low-res and unreadable. Figures need to be made much larger.

Changed

6) Line 305 spelling: “focusses”

We assume the reviewer meant the mis-spelling of “focussed”. Changed to focused rather than focussed

7) The classes of antibiotics chosen to be displayed in Fig 3 are odd and not comprehensive or reflective of antibiotic use patterns or previously reported abundant ARG classes. Beta-lactamases, one of the most common and widely disseminated classes of ARGs, is not mentioned in this paper.

We chose to label the top ten most abundant ARG classes and label the rest as “Other” rather than labelling based on the most disseminated antibiotic classes. As such, our results represent what is most commonly present, rather than what is assumed based on antibiotic use.

8) Line 159: it appears “containments” should be “contaminants”.

Paragraph now removed.

9) The language needs to be cleaned up in the paragraph on differential analysis (lines 205 – 227). The purpose of this analysis is not to determine which ARGs each sample “contains” but rather the ARGs that are differentially “enriched” in each sample type. Sample composition was already covered in the first paragraph of the results section.

Paragraph has now been modified accordingly.

10) Line 407: it appears “where” should be “were”.

Changed

11) Fig. 5; Supplementary Fig. 9 captions: “Bonferroni-Hochberg” should either be Bonferroni or Benjamini-Hochberg.

Changed

12) Supplementary figure numbering in the text appears to be wrong in some instances.

a) Lines 189 and 192: “Supplementary Fig. 5” appears to refer to Supplementary Fig. 4.

b) Line 207 “Supplementary Fig. 7” appears to refer to Supplementary Fig. 6.

c) Supplementary Fig. 7 is not referenced in the text.

Rectified

13) When multiple panels showing the same measurement are included in the same panel the y-axis should be the same to avoid misrepresentation of the data (Figs. 3a, 4; Supplementary Figs. 3, 6, 8.)

Changed

Figure 1: Relative abundance of ARG classes from results using tools Bowtie2, KMA, KMA and Bowtie2, and SRST2 with a 90% coverage threshold

Figure 2: Benchmarking of coverage threshold (red dashed line at 90 %) showing the distribution of the percentage coverage of ARGs using the KMA-Bowtie2 pipeline for all samples ($n = 1174$). Distributions are labelled by whether the equivalent ARG is present in (red) or absent from (blue) the metagenomic assemblies. Metagenomes were assembled using SPAdes 3.9.0 with parameters `-k 21,33,55 -only-assembler -meta`. These assemblies were aligned against CARD 3.0.0 using BLASTN 2.7.1 with parameters `-outfmt 6 -evalue 1e-10`. All hits were filtered by an e-value $\leq 1e-50$ and an identity ≥ 80 . For query sequence hits that overlapped greater than 20% of the smallest query sequence hit, hits were selected that had the lowest e-value, or the highest identity for hits with the same e-values. Other hits were discarded. We then make the assumption that an ARG in a sample should be found in at least one contig in an assembly. In reality, this may not be the case for ARGs with a low abundance or for lower quality assemblies. Considering this limitation, we use the presence or absence of ARGs in a metagenomic assembly as a proxy for ARG incidence in a sample.

Reviewer #2:

Methods.

Public available shotgun metagenomes were analysed; however, the sampling design of the studies analysed was different and the samples are very heterogeneous. In some cases, low number of individuals representing one country, e.g. five from Germany.

This is an important issue, and one which frequently affects metagenomic studies currently due to the costs of sequencing. We are aware of this and we only compare high-level incidence rather than abundance and diversity metrics across countries in Figure 1 and Figure 2a,b. Due to this issue, throughout our analysis we are careful to avoid statistical tests across cohorts and instead focus on statistical comparisons between paired body sites within cohorts. As such, differences across body-sites within the same individual, have been subjected to the same methodological/sequencing techniques which aids in comparability.

It is briefly mentioned that the quality of all datasets was checked following the standardized filtering. However, there is no information available on the quality requirements that were used for this study. Were datasets with lower quality removed from the analysis or manually filtered in order to meet the requirements?

Thank you for highlighting this significant omission. We have now added a sentence “All metagenomes passed over half the quality control metrics in FastQC (<https://www.bioinformatics.babraham.ac.uk/projects/fastqc/>) with these pass rates calculated in MultiQC” under “Metagenomic sequence data” heading in the Methods section (Lines 453 – 455 revised manuscript).

Only one method was applied: in silico analyses of metagenomes. For risk assessments of antibiotic resistances it is much better to combine various methods and set everything in a specific context.

We agree that combination-based approaches provide a powerful mechanism for elucidating biological events etc. However, this study used a data-driven approach rather than an approach targeted at a particular antibiotic or clinical question. Sufficient metadata for usable data sets does not exist on previous antimicrobial use per individual for a thorough analysis of a context-specific study.

The authors mention (when referring to previous studies) that individuals that were not exposed to anthropogenic sources of therapeutic antibiotics also harbor diverse ARGs in their microbiome. This is later on mostly ignored when the authors draw conclusions from their own results.

This oversight has now been corrected, with further debate added to the Discussion section.

In order to better understand the relevance of the findings it would be necessary to correlate the data with more precise metadata, which indicates which individuals were exposed to certain therapeutic antibiotics.

This would be the absolute ideal and gold standard which we would be ecstatic to be able to do. However, as we have used open-source metagenomic data from previous studies with different objectives, it is not possible to get complete, consistent metadata for all the different cohorts. We have sacrificed complete metadata for broader population coverage. However, future studies on discrete populations will hopefully be undertaken that will contain more precise and relevant metadata as the nascent field of metagenomics and resistome progresses. This is something which we look forward to in great anticipation!

The metagenomic data should be ideally linked to other -omics data (metagenomics or – transcriptomics) to assess the expression of ARGs, otherwise it can't be excluded that inactive artifacts are assessed.

Whilst metatranscriptomics data for these cohorts would, indeed, be fascinating and informative about a broad range of microbiome-related factors, it would not be particularly informative about antimicrobial resistance unfortunately. This is predominantly due to the fact that most ARGs are generally inactive and only expressed when the host microbe is undergoing external antimicrobial selection pressures (i.e. the host is taking antibiotics). As one of the most consistent exclusion criteria for microbiome studies is that the subjects not be taking any antimicrobial drugs, as well as not having taken them for anywhere between 1 – 3 months prior to the sampling, this means that none of these patients is likely to be under current antimicrobial selection pressure. Further, it is unlikely in the extreme that any cohort subject would ever undergo treatment with more than 2 or 3 antibiotics at a time. Thus, transcriptomics are unlikely to detect many (if any) of these inducible genes in these untreated cohort subjects. This is true of any other omics approach designed to detect current expression/production of any of these genes, including proteomics and metabolomics. A better approach that is gaining increasing interest and that we are considering for future studies is the use of Functional Metagenomics to determine which of the ARGs identified as capable of being expressed when a host microorganism is placed under antibiotic selection pressure.

Reviewer #3:

My only concern is whether the data from the different studies are comparable. Thus, we do know that different DNA-purification methods can drastically change the composition of the microbiome, including the resistome. Thus, some evidence that the differences observed is not only due to the data generation would be nice.

We would like to thank the reviewer for addressing this important issue. We have answered this under Reviewer 1's comment 8).

I am a bit surprised with the correlation between antimicrobial use and resistance. This is in some contrast to recent studies looking at both metagenomics and single isolates (see Collignon. lancet Planetary health and hendriksen Nature Commun.).

We were also surprised by this finding. In light of this, we did not feel our analysis in the original manuscript was adequate enough as it used the proportion of samples that contained an ARG class. This meant if every individual had just one ARG from a particular class, the proportion would be 100% which we felt could overestimate resistance potential. In our revised manuscript, we have opted to use the mean abundance of ARG classes for each individual in a repeated analysis. However, again, we do not find any statistical linear correlation between mean ARG class abundance and prescription rates.

I only have some problems identifying the real novelty of the study and the purpose of doing it. The main findings seem to be that the oral cavity contains more resistance and less diversity compared to gut microbiome and then that some abundances of bacteriophages are associated to some resistance genes.

We would like to thank the reviewer for raising the novelty and purpose of this study. We did not make this sufficiently clear in our manuscript.

With regards to the novelty of this particular study, to our knowledge, no worldwide human resistome studies using shotgun metagenomic data have been applied to different oral sites and comprehensively compared with other body sites. We know of two smaller studies that used functional or targeted rather than shotgun metagenomics to look for particular tetracycline and erythromycin ARGs rather than the resistome in the oral cavity, but did not compare with other body sites (Diaz-Torres et al., 2006; Seville et al., 2009). Another smaller

study with 34 participants used shotgun metagenomics to compare resistance classes between oral mucosa and stool, but not compare ARGs (Clemente et al., 2015).

Our research expands from these to characterise the resistome (the total ARGs) across different oral sites and compares this with paired gut samples using publicly available shotgun metagenomic data across different countries. As well as comparing ARG classes and mechanisms across body sites, we compare ARG class abundance with prescription rates, compare ARG prevalence (is it present or absent), ARG abundance (how much of each ARG), ARG diversity (how many different types), predict the bacterial hosts of ARGs and compare stability of ARGs across longitudinal US samples.

In terms the purpose of further investigating the oral cavity, most human resistome studies to date use gut as a model representation of potential resistance (Forslund et al., 2013; Palleja et al., 2018; Pärnänen et al., 2018). The gut contains a reservoir of ARGs, likely due to the presence of hosts that carry many ARGs like *E. coli* and higher exposure and selection pressures from orally administered antimicrobials. Although the gut contains diverse ARGs, their role in potential resistance in different body sites is unclear. ARGs in the gut have low abundance and the potential for expression to drive resistance in other body sites are not known. By investigating ARG profiles in oral sites with different microbes and exposure to antibiotics, we find fewer ARGs, but at higher abundances. We hypothesise that ARGs have been selected in the oral cavity by pressures of either a lower exposure of antibiotics entering the bloodstream from the gut or topical antibiotics for oral infections. If some ARGs were indeed selected by a lower exposure to circulating antibiotics and are found in higher abundances, these ARGs have the potential to become more resistant to further exposure to the same or similar antibiotics.

Therefore, we suggest different body sites have different roles in persistence and prevalence of ARGs depending on their environment and antibiotic exposure, which is likely to affect their resistance potential and their relevance at particular body sites. (Discussed in more detail in Lines 430 – 440 of revised manuscript).

Considering the very large number of resistance genes they looked at the latter is not surprising and it would have been nice with some verification using long-read sequencing (I know that would require new samples) or at least binning to show co-localisation.

For benchmarking our predictions, we used a validation method based on comparing the incidence of ARGs on assembled contigs. This is described in our response to question 2) from Reviewer 1.

Long-read sequencing technologies (i.e. PacBio or Nanopore) could be useful for supporting our predictions. One problem with using PacBio is that it relies on having a high amount of DNA from pooled samples. This amount of microbial DNA is especially difficult to extract from saliva and buccal mucosa with a low percentage of microbial DNA. Nanopore sequencers do not require as much DNA, but it has lower accuracy with complex metagenomic samples. More importantly, Nanopore extraction protocols for oral (saliva) samples are still under development, but we are eager to apply Nanopore to saliva samples in the future.

Binning to co-localise ARGs with taxonomic units or Metagenomic Species Pan-genomes could be used to predict the host origin of these ARGs. There are many binning tools available that bin contigs by sequence features, such as GC content, k-mer frequency etc. However, different sample types will have varying completeness of bins because of differences in sequencing depths. Thus, it would be difficult to compare predictions across sample types. Therefore, on this occasion, we chose to use co-association of ARG and

species abundances using Spearman's rank with a strict criteria (described Lines 616 – 619 in revised manuscript).

References

Clausen, P.T.L.C., Zankari, E., Aarestrup, F.M., and Lund, O. (2016). Benchmarking of methods for identification of antimicrobial resistance genes in bacterial whole genome data. *J. Antimicrob. Chemother.* *71*, 2484–2488.

Clausen, P.T.L.C., Aarestrup, F.M., and Lund, O. (2018). Rapid and precise alignment of raw reads against redundant databases with KMA. *BMC Bioinformatics* *19*, 307.

Clemente, J.C., Pehrsson, E.C., Blaser, M.J., Sandhu, K., Gao, Z., Wang, B., Magris, M., Hidalgo, G., Contreras, M., Noya-Alarcón, Ó., et al. (2015). The microbiome of uncontacted Amerindians. *Science Advances* *1*, e1500183.

Diaz-Torres, M.L., Villedieu, A., Hunt, N., McNab, R., Spratt, D.A., Allan, E., Mullany, P., and Wilson, M. (2006). Determining the antibiotic resistance potential of the indigenous oral microbiota of humans using a metagenomic approach. *FEMS Microbiol Lett* *258*, 257–262.

Forslund, K., Sunagawa, S., Kultima, J.R., Mende, D.R., Arumugam, M., Tydas, A., and Bork, P. (2013). Country-specific antibiotic use practices impact the human gut resistome. *Genome Res.* *23*, 1163–1169.

Hendriksen, R.S., Munk, P., Njage, P., Bunnik, B. van, McNally, L., Lukjancenko, O., Röder, T., Nieuwenhuijse, D., Pedersen, S.K., Kjeldgaard, J., et al. (2019). Global monitoring of antimicrobial resistance based on metagenomics analyses of urban sewage. *Nature Communications* *10*, 1124.

Hunt, M., Mather, A.E., Sánchez-Busó, L., Page, A.J., Parkhill, J., Keane, J.A., and Harris, S.R. (2017). ARIBA: rapid antimicrobial resistance genotyping directly from sequencing reads. *Microbial Genomics* *3*.

Palleja, A., Mikkelsen, K.H., Forslund, S.K., Kashani, A., Allin, K.H., Nielsen, T., Hansen, T.H., Liang, S., Feng, Q., Zhang, C., et al. (2018). Recovery of gut microbiota of healthy adults following antibiotic exposure. *Nature Microbiology* *3*, 1255.

Pärnänen, K., Karkman, A., Hultman, J., Lyra, C., Bengtsson-Palme, J., Larsson, D.G.J., Rautava, S., Isolauri, E., Salminen, S., Kumar, H., et al. (2018). Maternal gut and breast milk microbiota affect infant gut antibiotic resistome and mobile genetic elements. *Nat Commun* *9*.

Seville, L.A., Patterson, A.J., Scott, K.P., Mullany, P., Quail, M.A., Parkhill, J., Ready, D., Wilson, M., Spratt, D., and Roberts, A.P. (2009). Distribution of tetracycline and erythromycin resistance genes among human oral and fecal metagenomic DNA. *Microb. Drug Resist.* *15*, 159–166.

Sturød, K., Dhariwal, A., Dahle, U.R., Vestrheim, D.F., and Petersen, F.C. (2019). Impact of narrow spectrum Penicillin V on the oral and fecal resistome in a young child treated for otitis media (*Microbiology*).

Zaheer, R., Noyes, N., Polo, R.O., Cook, S.R., Marinier, E., Domselaar, G.V., Belk, K.E., Morley, P.S., and McAllister, T.A. (2018). Impact of sequencing depth on the characterization of the microbiome and resistome. *Scientific Reports* 8, 5890.

REVIEWERS' COMMENTS:

Reviewer #2 (Remarks to the Author):

The manuscript has been substantially improved.

Reviewer #3 (Remarks to the Author):

I do not really have anything further to add to my previous review.

The inherent problems by comparing different studie is not really solved and the authors do not really provide anything to solve it, except stating that they only look at resistance at a higher level. One could have expected that they compared the complete composition using minHash clustering and thus, provided some kind of evidence that the overall composition was sufficient similar to allow comparisons at the AMR-gene level.

Reviewer #4 (Remarks to the Author):

The authors have done a good job of responding to the prior comments from Reviewer 1.

I have offered a few additional minor comments, below:

- A personal style issue – I find it a bit sensationalistic to include the phrase “antimicrobial resistance apocalypse” in the first sentence of the abstract.

- It is interesting that the oral microbiome resistome does not correlate with antibiotic prescription rates. As triclosan containing toothpaste has been demonstrated to correlate with alterations in the gut microbiome (PMID: 29030459), do the authors have information on the use of microbicide-containing toothpastes among the studied subjects? I realize that this information may not be publicly available in the metadata associated with these published datasets, but it is worth making a point about in the discussion.

- The wording for lines 31-32 is awkward. I am not sure what is meant by “the highest and lowest abundance of specific antimicrobial resistance genes”.

- Line 56 onward - The authors state that commensal microbes from the oral cavity have considerable potential to lead to AMR infections at other body sites. I would delete the word “considerable”, as what constitutes “considerable” is arguable.

- Line 141 – wording of “mechanisms of reduced permeability to antibiotic” is awkward.

- Line 168 – I find the terms “penams, penems...” to be a bit nonstandard. While the terms are absolutely correct, they are not in common usage. Thus, it might be preferable to explain them in a more detailed manner.

- Line 216 – would explain what is meant by a “resistotype” in this context.

- Read length correction – The authors note (in the response to reviewer 1’s comments) that all samples reported in this study were sequenced on a Hiseq 2000 – however, different kits can produce different read lengths.

- Line 485 – I’m not sure “extracting” is the correct term for what was done. Perhaps “identifying” or “quantifying” is more accurate?

Reviewer #3 (Remarks to the Author):

I do not really have anything further to add to my previous review.

The inherent problems by comparing different studies is not really solved and the authors do not really provide anything to solve it, except stating that they only look at resistance at a higher level. One could have expected that they compared the complete composition using minHash clustering and thus, provided some kind of evidence that the overall composition was sufficiently similar to allow comparisons at the AMR-gene level.

The reviewer makes a good suggestion of using minHash to find whether the composition is sufficiently similar across cohorts. In fact, at the start of the work for this study, we used a minHash clustering algorithm on k-mer composition (<https://github.com/GATB/simka>) on the China cohort only to check whether the body site labels provided by the authors were correct. However, at the time, we did not apply this to other cohorts.

When we applied Metaphlan2 to quantify the microbial composition of every sample, we found the microbial compositions clustered by body site and not so much by cohort. There was some deviation between US and China stool samples that may be attributable to differences in diet and lifestyle. However, the metadata from these studies does not allow us to interrogate this in greater depth.

Reviewer #4 (Remarks to the Author):

The authors have done a good job of responding to the prior comments from Reviewer 1.

I have offered a few additional minor comments, below:

- A personal style issue – I find it a bit sensationalistic to include the phrase “antimicrobial resistance apocalypse” in the first sentence of the abstract.

We agree with the reviewer that this sounds too sensationalist. The word “apocalypse” and quotations have been removed.

- It is interesting that the oral microbiome resistome does not correlate with antibiotic prescription rates. As triclosan containing toothpaste has been demonstrated to correlate with alterations in the gut microbiome (PMID: 29030459), do the authors have information on the use of microbicide-containing toothpastes among the studied subjects? I realize that this information may not be publicly available in the metadata associated with these published datasets, but it is worth making a point about in the discussion.

We strongly agree that if these data were available, they would contribute to an important study into the effects of use and exposure to triclosan in mainstream household products. Unfortunately, triclosan use data were not publicly available for any country or region of the datasets we used in this study.

As shown in the manuscript, we do find a high proportion of stool samples from China, Fiji and Western Europe contain triclosan resistance, but interestingly not the US. This observation has been added to the manuscript (Lines 173 - 175). The FDA in the US ban household cleaning products containing triclosan, like toothpaste. By way of comparison, the

EU regulates the use of triclosan and so it is unregulated in the Philippines and Fiji. Thus the resistance patterns identified here could be related to region-wide differences in triclosan regulation, but this is further complicated by the historical use of triclosan and use of specific products containing triclosan. Further, without the specific use data on these subjects, this is speculative at best.

- The wording for lines 31-32 is awkward. I am not sure what is meant by “the highest and lowest abundance of specific antimicrobial resistance genes”.

Some ARGs that have a very low abundance are also found in the oral cavity as well, but this is also true in the gut as well. We agree this is confusing and does not add any insight into our main findings. The sentence has now been changed to “highest abundances of antimicrobial resistance genes are found in the oral cavity”.

- Line 56 onward - The authors state that commensal microbes from the oral cavity have considerable potential to lead to AMR infections at other body sites. I would delete the word “considerable”, as what constitutes “considerable” is arguable.

Removed “considerable”

- Line 141 – wording of “mechanisms of reduced permeability to antibiotic” is awkward.

Changed to “Reduced permeability to antibiotics” without the word “mechanism”.

- Line 168 – I find the terms “penams, penems...” to be a bit nonstandard. While the terms are absolutely correct, they are not in common usage. Thus, it might be preferable to explain them in a more detailed manner.

Penam has been defined as a “β-lactam with saturated five-membered ring such as penicillin” and penems as a “β-lactam with unsaturated five-membered ring” in the first mention of penam and penem in Lines 170-171.

- Line 216 – would explain what is meant by a “resistotype” in this context.

Added “clustered into distinct groups termed “resistotypes”

- Read length correction – The authors note (in the response to reviewer 1’s comments) that all samples reported in this study were sequenced on a Hiseq 2000 – however, different kits can produce different read lengths.

The reviewer makes a very good point about differences in read length. To measure ARG abundance, we normalised the number of reads aligned to an ARG (read count) using RPKM. Differences in read length would have little impact on RPKM.

RPKM is defined as:

$RPKM = \text{read count} / (\text{total reads mapped in millions} * \text{gene length in kilobases})$

For example, if 1 million reads of sample were sequenced with an average length of 150 nt, and an ARG length of 1000 nt had a raw read count of 50, then the RPKM would be 1 ($50/1*1$). If these reads had an average length of half (75 nt), then 2 million reads would be

sequenced (assuming sequencer is not limited) and double the number of reads (100 reads) are expected to align to the ARG. The RPKM ends up being 50 also ($100/2*1$).

- Line 485 – I'm not sure "extracting" is the correct term for what was done. Perhaps "identifying" or "quantifying" is more accurate?

We agree the term "extracting" can be confused with laboratory extraction. All instances of "extract" in the manuscript have been replaced by "identify" or "quantify" apart from "extraction protocols".